# Socioeconomic Determinants of Households' Vulnerability to Drought in Western Cape, South Africa

**Isaac Busayo Oluwatayo** [1,*] and **Tamunotonye Mayowa Braide** [2]

1   Department of Agricultural Economics and Agribusiness, University of Venda,
   Thohoyandou 0950, South Africa
2   Department of Agricultural Economics and Animal Production, University of Limpopo,
   Sovenga 0727, South Africa; braidetamunotonye@gmail.com
*   Correspondence: isaac.oluwatayo@univen.ac.za

**Abstract:** This paper examines the determinants of household vulnerability to drought in the Western Cape province, South Africa. The study used secondary data collected by the Human Sciences Research Council (HSRC). The dataset is made up of 240 households located in the City of Cape Town along with two small towns in the West Coast District Municipality (Piketberg and Clanwilliam). While descriptive statistics were used to analyse households' socioeconomic variables, an ordered logit model was employed to analyse the factors contributing to households' vulnerability to drought in the study area. The paper revealed that 28% of the households were extremely vulnerable to drought. The result of the ordered logit regression model showed that factors such as the age of the household head, communication of water restrictions by the authorities, household water consumption in the last two years, and public cooperation with water restrictions were significant factors influencing households' vulnerability to drought. It was also discovered that female-headed and older household members were more vulnerable to drought than their male-headed and younger members respectively. The paper concluded that to minimise vulnerability to drought among the households, stakeholders in the province should be better prepared to implement proactive policies with regard to climate disasters.

**Keywords:** drought; ordered logistic regression; vulnerability; Western Cape province

## 1. Introduction

Drought is a complex natural hazard that affects large areas over an extended period of time and has devastating effects on water supply, crop yield, and the environment [1,2]. It is a significant water deficit caused by climatic factors such as decreased rainfall or human factors such as land-use change [3]. Globally, drought is becoming more severe as the areas affected by drought keep increasing [4]. It has become a growing concern in many parts of the world as the population and businesses grow [5–8]. Climate change has also caused climate zones to shift in many parts of the world, with dry areas increasing and Arctic areas decreasing [4].

Droughts are common occurrences in South Africa. However, there has been an increase in multi-year droughts in recent years. For example, the Western Cape province, where the study area is located, was declared a disaster area following a severe drought that occurred between 2015 and 2018. [9–11]. The consequence of drought began when the level of the dam fell to 71% in 2015, 60% in 2016 and 38% in 2017 [12]. Water dams are regarded as the most valuable and efficient man-made water storage facilities to manage water resources from a socioeconomic standpoint [13,14].

The frequency and severity of droughts in recent years have posed challenges to the socio-economic development of developing countries, particularly in the agricultural sector [15]. Drought affects the socioeconomic status of affected households and

communities [4]. The drought in the Western Cape province of South Africa affected the highly populated area of Cape Town and its water supply [16]. Increased water demand, insufficient water supply planning and management, insufficient investment in water reservoir infrastructure, and recurring droughts have put the city and surrounding communities under acute water protection strain over the last decade [17]. The authorities' and agencies' responses to the drought in the study area showed a lack of planning for the event, causing vulnerability among the households in the area. Thus, it is important to investigate the impact of natural disasters and how the respondents understand and interpret the disaster to reduce the vulnerability of households. This is important to assist policymakers in formulating appropriate policies to prevent, rather than merely reactive to these disasters. Therefore, this study investigates the determinants of household vulnerability to drought in the Western Cape province, South Africa.

## 2. Literature Review

### 2.1. Drought

The concept of drought is determined by the perspectives and needs of those who are affected. It varies from short dry spells lasting from a few weeks to several months, years, or even decades, and can occur in even the most humid parts of the world [18]. Droughts should be distinguished from aridity and the regular dry seasons found in tropical and subtropical regions around the world. It is classified according to its frequency, intensity, duration, and extent [19,20].

Drought is classified into four categories based on its impact: meteorological, hydrological, agricultural, and socioeconomic droughts [21]. According to existing classification [22], these drought types tend to occur in the following order: climate variability induces a precipitation shortage, which causes a meteorological drought, which, when combined with high potential evapotranspiration, leads to agricultural or soil moisture drought. The effects of temperature fluctuations, precipitation shortfalls, and anthropogenic demand pressures on surface or subsurface water supply, such as streams, reservoirs, lakes, or groundwater, cause hydrological droughts. While the socioeconomic drought is associated with the impact of an inadequate supply of water resulting from meteorological, agricultural, and hydrological droughts [22,23]. Socioeconomic drought is a type of water scarcity produced by an imbalance between the availability of water resources in natural systems and the demand for water in human socioeconomic systems [24]. Even if a socioeconomic drought is over, the antecedent water shortage may have long-term consequences, altering the resilience of a regional water resource system [25].

### 2.2. Impact and Social Aspects of Drought

Droughts are frequent, slow-onset threats that can have substantial direct and indirect impacts on humans and the environment [26]. Prolonged drought spells are often the result of a combination of natural and social factors [27]. Drought can have an economic, environmental, or social impact. Drought has social implications such as water shortages, health issues that end in death, forced migration, conflicts, and hunger/famine, as well as economic consequences such as lost income/livelihoods and competition for decreasing resources [4]. Environmental impacts include forest fires, tree mortality, land degradation, loss of ecosystem functioning, reduced carbon sequestration, and disrupted carbon cycles [4]. It also affects agriculture by lowering the quantity of water accessible to cattle and crops. Drought mortality is highest in sub-Saharan Africa, while drought economic losses are highest in more developed nations such as Western and Southern Europe, North and Central America, the Middle East, Australia, and north-eastern China [28–30]. The potential for negative consequences is created by a combination of hazards, exposure, and vulnerability, rather than by natural disasters (droughts, floods, etc.).

The international community, notably UN agencies, examined the link between drought, water scarcity, and poverty. The Sustainable Development Goal (SDG) 6, "Ensure access to water and sanitation for all" [31,32], now addresses it, building on the previous

Millennium Development Goal (MDG) 7 and its sub-target 10, which aimed to reduce "the proportion of people without sustainable access to safe drinking water and basic sanitation by 2015." [33]. Drought, lack of access to water, and inadequate freshwater circulation all undoubtedly contribute to other issues addressed by the development goals such as food security (MDG 1, SDG 1, and 2) [34]. Over the last few decades, agricultural expansion has reduced agricultural systems' abilities to cope with drought in semi-arid and dry sub-humid tropical areas [35]. Drought in some of the world's most important food-producing areas has resulted in an increase in food costs due to the globalization of the food supply and processing network [36].

Vulnerability, resilience, and adaptation are used to analyze water scarcity and droughts, as well as their socio-ecological implications [37]. These concepts are interconnected, and their existence varies across spatial scales within various ecosystems and societies. A society's vulnerability is defined not only by its physical environment and the occurrence of natural hazards in that environment, but also by the diverse economic, political, and social characteristics of that society [38]. Vulnerability has three components: exposure, sensitivity, and adaptive capability. Resilience is defined as a system's capacity to adjust and sustain disruptions and shocks while staying in the same condition, whereas adaptation is defined as the transformation of natural or human systems in response to present or expected climatic stimuli or their impacts [39,40].

### 2.3. Drought in Western Cape Province

The Western Cape province is located in the southernmost part of the African continent, as shown in the right bottom corner of Figure 1 [41]. The Western Cape province of South Africa suffered from severe drought, which had a particularly negative impact on the densely populated area of Cape Town and its water supply (2015–2017) [16]. An El Niño-triggered drought has affected South Africa's Western Cape since 2015 [16]. As shown in Figure 1, the drought was caused by a prolonged below-average monthly rainfall sequence that began in 2015, with some regions of the Western Cape having received no rain since 2014, and it intensified throughout the rainy season of April to September 2017 [42,43]. When the water crisis started in the Western Cape in 2015, extending to 2017 and 2018, it placed enormous pressure on the City of Cape Town's (CoCT) water supply. The water supply for the region is fed by six dams in the catchment and supplies a large agricultural area and a number of municipalities, of which the Cape Town metropolitan area, with approximately four million Cape Town inhabitants, is one [44–47]. In response to the intensification of the drought, CoCT gradually undertook several actions to reduce water use. Cape Town's water supply was in critical condition, and water for any use was limited by water cuts and rationing [12]. A communication campaign took place through radio, print, and social media, essentially reaching out to citizens and mobilising for the reduction in the city's water consumption, as well as encouraging people to use less water and stay below the water restrictions [47]. The drought and the associated water crisis have had large socio-economic impacts at national and local levels. The Western Cape province contributed about 14% to the country's gross domestic product and its agricultural sector, one of the most affected because of the high dependence on water for irrigation, reported losses estimated at ZAR 5.9 billion in the 2017–2018 season [9].

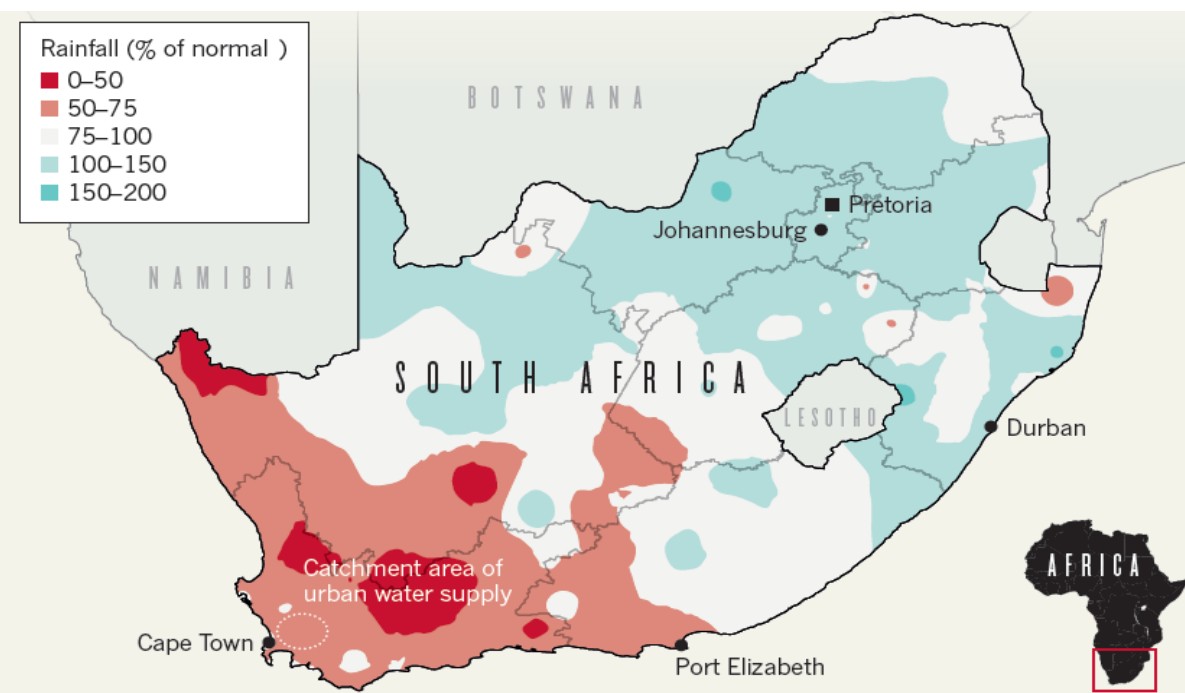

**Figure 1.** The map of Western Cape province in South Africa and rainfall distribution [41].

### 2.4. Water Sustainability in South Africa

South Africa is a water-scarce country. In recent years, water security has come to prominence in international policy debates, drawing attention to political, economic, and social determinants of access to safe and sufficient water, as well as hydrological and climatic factors [48]. Sustainable water systems are presented as the way to avoid crises of water shortages and floods in cities, and the related concept of resilience enables sustainable cities to respond more effectively to extreme events, which are more likely in an uncertain future [49]. Sustainable access to improved water service is defined as a daily supply of portable water that is sufficient, affordable, dependable, and continuous [50]. Water scarcity implies the need for additional water sources, for example, seawater desalination, stormwater harvesting, wastewater reclamation, and so on. Wastewater reclamation is the process of reclaiming wastewater into a reusable form through artificial treatments. Through the treatment processes, the reclaimed water can meet appropriate water quality criteria so it can be returned to the environment to augment the natural systems from which it came or be reused for numerous purposes related to human activities [51]. The process of removing excess salts and other dissolved chemicals from seawater is known as desalination [52]. It is capital-intensive and requires a substantial amount of energy from nonconventional energy sources, which are unsustainable. Renewable energy sources such as solar, wind, and geothermal energy can be used to power desalination processes [53]. The largest desalination plant in South Africa supplies 15 million litres per day and is in Mossel Bay. The CoCT has undertaken feasibility studies to investigate whether sea water desalination is viable at a large scale (more than 100 Ml/day) [54].

### 3. Methodology

#### 3.1. Study Area

The Western Cape province in South Africa is the country's southernmost province, covering an area of more than a million square kilometers [43,44,54]. The Western Cape (WC) is one of South Africa's nine provinces, situated on the southwest coast of the country with a long coastline. The province is made up of twenty-five municipalities grouped into six districts. The borders of these municipalities are shown in Figure 2. The Western Cape is the fourth largest of the nine provinces of South Africa. On the coasts, the climate is mildly

Mediterranean (wet winter and dry summer), transitioning to semi-arid and continental as one travels inland and past the mountains. The vegetation is mostly scrubland, a common and fire-prone vegetation type [43]. Agriculture is a growing industry in the area, which is home to renowned vineyards [54]. The Western Cape is home to one of the country's capitals, Cape Town, as well as one of the major urban areas in South Africa. Most of the province's population resides in the Cape Town metropole area (64%) [55]. The population estimate for 2019 was 6,844,272 inhabitants [56]. The drought was reported to be having severe impacts on the population, with the water level of the Theewaterskloof dam (the largest dam in the Western Cape water supply system, holding 41% of the water storage capacity available to Cape Town) critically low [57]. This study was conducted among households in the City of Cape Town (CoCT) and the West Coast District Municipality (Clanwilliam and Piketberg).

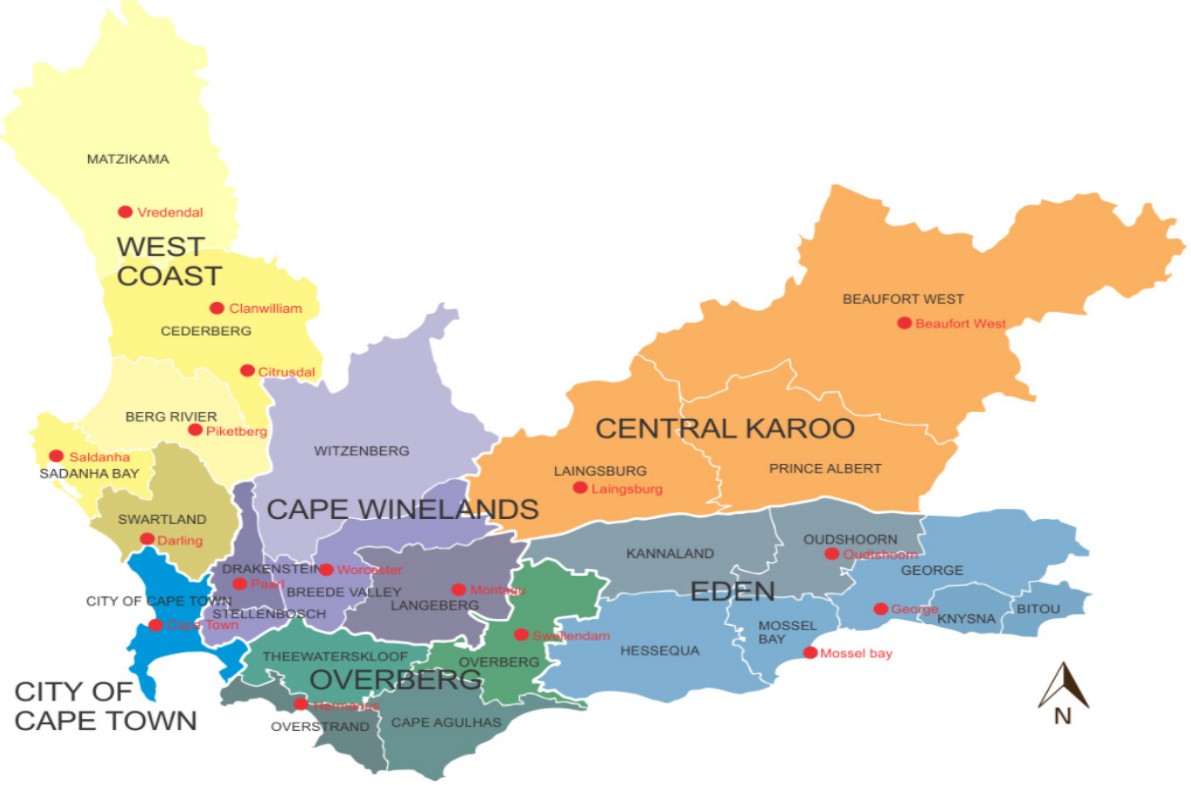

**Figure 2.** Map of Western Cape province, South Africa [58].

*3.2. Data Sources and Sampling Method*

This study used secondary data collected by the Human Sciences Research Council (HSRC) [58]. This study focused on households. Within each of the selected areas, cluster samples were selected, to get a minimum of twenty households per area. In most cases, the targets were exceeded. The data focused on respondents' behaviour and perceptions during the period of severe drought in the Western Cape from 2016 to 2018. According to HSRC [58], the origin of the data is a combination of primary and secondary data. Seven areas were selected to represent the diverse races and socio-economic circumstances of the Western Cape population. Five of the localities are situated in the City of Cape Town (CoCT), and two are small towns in the West Coast District Municipality (Piketberg and Clanwilliam). It consists of 240 households located within each of the seven selected sampling areas [58].

*3.3. Analytical Methods*

Descriptive statistics, such as graphs and tables with frequencies and percentages, were used to describe the socio-economic characteristics of households as well as their

behaviour and awareness of saving water and water restrictions; communication of water restrictions and disasters by authority; the effect of drought on the household; public cooperation with water restrictions; and the water consumption of households in the selected study areas. The ordered logit model was used to analyse the determinants of the households' vulnerability to drought in the Western Cape province, South Africa.

Ordinal Logistic Regression (OLR)

In the context of Ordinal Logistic Regression (OLR), ordinal means the order of the categories. The OLR is, therefore, a regression technique used when the dependent variable is measured at the ordinal level and given one or more explanatory variables, which could be ordinal, continuous, or categorical [59]. The reason for analysis with OLR is that the dependent variable is categorical and ordered. The OLR considers the probability of that event and all others above it in the ordinal ranking [60]. We are concerned with cumulative probabilities rather than probabilities for discrete categories.

Hence, the model:

$$logit P((Y \leq j)) = \beta_{j=0} - \beta_{j=1} x_1 + + \cdots + \beta_{j=p} \, x_p \, for \, j = 1, \ldots, j - 1 \tag{1}$$

with $P$ predictors is called the ordinal logistic regression model.

In addition, let $\{p_0, p_1, \ldots, p_{j-1}\}$ be the associated probabilities. The cumulative probability of a response less than and equal to $j$ is given as:

$$P(Y \leq j) = \frac{\exp(\alpha j + \beta X)}{(1 + \exp(\alpha j + \beta X))} \tag{2}$$

where:

$$log\left(\frac{P(Y \leq j)}{P(Y > j)}\right) = \alpha j - \beta X, \, j \, \epsilon \, [1, \, J - 1] \tag{3}$$

and $\alpha j$ is the intercept and the log odds of falling into category $j$ or below.

$\beta_k$ is the parameter that describes the effect of the independent variable $X_i$ on the dependent variable $Y$.

The cumulative logit is given as:

$$log\left(\frac{P(Y \leq j)}{P(Y > j)}\right) = log\left(\frac{P(Y \leq j)}{1 - P(Y \leq j)}\right) = log \, \frac{p1 + \ldots + pj}{pj + 1 \ldots + pj} \tag{4}$$

The cumulative logit measures how likely the response is to be in category $j$ or below versus in a category higher than $j$.

The following are the main assumptions that the OLR makes about the underlying data [58]:

- The dependent variable is ordinal.
- One or more of the explanatory variables are either continuous, categorical, or ordinal.
- There is no multi-collinearity.
- The odds are proportional. This means that each independent variable has an identical effect at each cumulative split of the ordinal dependent variable.

The dependent variable and the independent variable are shown in Table 1. The dependent variable is "households' vulnerability to drought", which has four ranked levels—"no vulnerability," "low-level vulnerability," "mid-level vulnerability," and "high-level vulnerability." OLR considers the order and contribution information of each independent variable [61]. As a result, it was applied to investigate the socio-economic determinants of households' vulnerability to drought.

**Table 1.** Socioeconomic determinants of households' vulnerability to drought (description of variables).

| Variables | Description | Unit of Measurement | Expected Outcome |
|---|---|---|---|
| $Y_i$ | Household vulnerability to drought | 1—No vulnerability<br>2—Low-level vulnerability<br>3—Mid-level vulnerability<br>4—High-level vulnerability | |
| $X_1$ | Gender of household head | 1—Male<br>2—Female | $+/-$ |
| $X_2$ | Age of household head | 1—(18–29)<br>2—(30–49)<br>3—(50–69)<br>4—(70+) | $+/-$ |
| $X_3$ | Household size | | $+$ |
| $X_4$ | Household water consumption in the past two years | 1—Decreased<br>2—Stayed the same<br>3—Increased | $+/-$ |
| $X_5$ | Communication of water restriction by authorities | 1—Yes<br>2—Partly<br>3—No | $-$ |
| $X_6$ | Effectiveness of authority to drought disaster | 1—Strongly Agree<br>2—Agree<br>3—Neutral<br>4—Disagree<br>5—Strongly disagree | $-$ |
| $X_7$ | Public cooperation with water restrictions | 1—Strongly Agree<br>2—Agree<br>3—Neutral<br>4—Disagree<br>5—Strongly disagree | - |
| $X_8$ | Household water-saving methods | 1—Did not save water<br>2—Used less water<br>3—Recycling<br>4—Fixed leakages<br>5—Stored water | $-$ |

## 4. Results and Discussion

### 4.1. Socioeconomic Characteristics of Households in Selected Areas of Western Cape Province, South Africa

Figure 3 shows the distribution of the households' vulnerability to drought in the study area. The result shows that at least 6% of the households in the study area were not vulnerable to drought during the period. In all, about 94% of the households in the study area were vulnerable to drought during the period, with about 33%, 33% and 28% of the households experiencing low-level, mid-level and high-level vulnerability to drought, respectively. This finding is in line with research conducted in Mopani District Municipality in Limpopo Province, South Africa, which found that the majority of respondents were drought vulnerable, with only a few households experiencing low vulnerability [62]. This finding indicates that households in drought-affected areas are more vulnerable to drought.

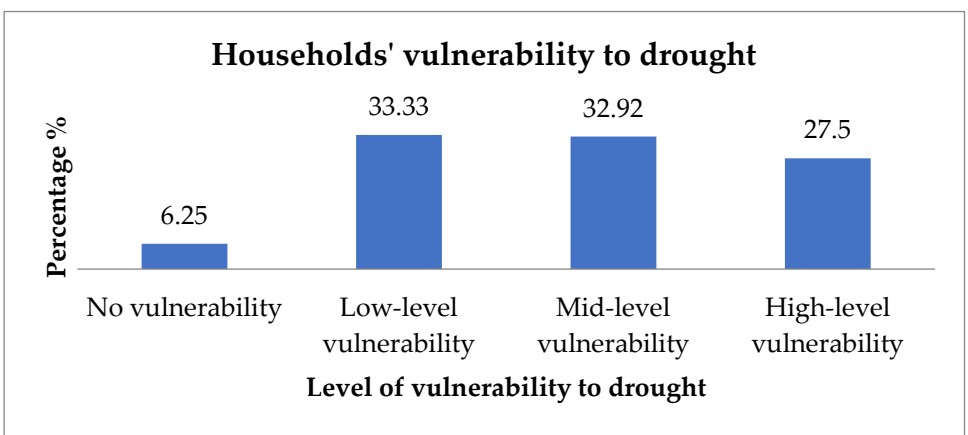

**Figure 3.** Households' vulnerability to drought [58]. **Source:** Authors' computation from survey data.

Figure 4 shows the distribution of respondents by location in the Western Cape province. The result shows that about 64% of the households were in CoCT with the following areas, such as CoCT-Township, CoCT -East, CoCT-Central, CoCT-South and CoCT-South, represented at 25%, 13%, 10%, 8% and 7%, respectively. Clanwilliam and Piketberg were also represented at 16% and 20%, respectively, in the West Coast District Municipality.

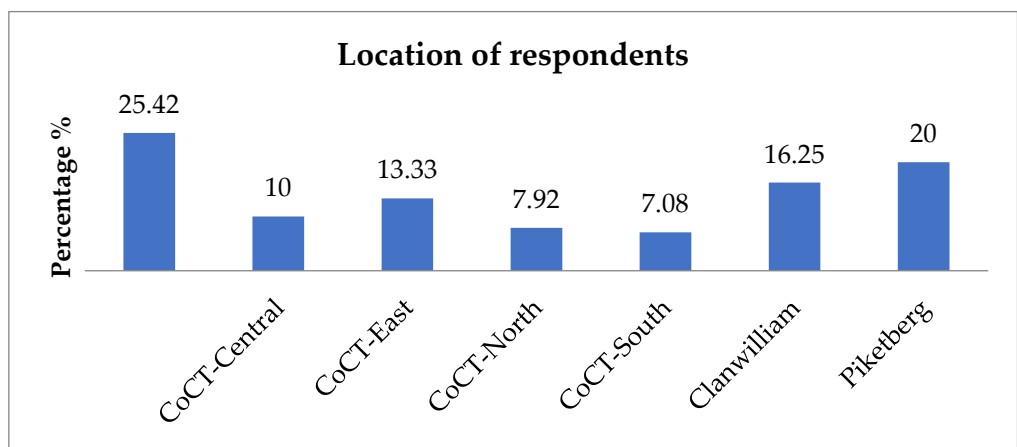

**Figure 4.** Location of respondents/heads of households [58]. **Source:** Authors' computation from survey data.

Figure 5 shows the distribution of households by sex in the study area in Western Cape province. The result shows that about 60% of the household heads were female, while about 40% were male-headed households. Less than 1% of the household heads represented those that were either non-specified or of different sexualities. According to Statistics South Africa [56] and the Western Cape government [63], there were more females (2,321,185 and 233,909) than males (2,277,600 and 229,484) in Cape Town and West Coast districts, respectively. This is consistent with the study as there are more female-headed households among the selected respondents in the study area.

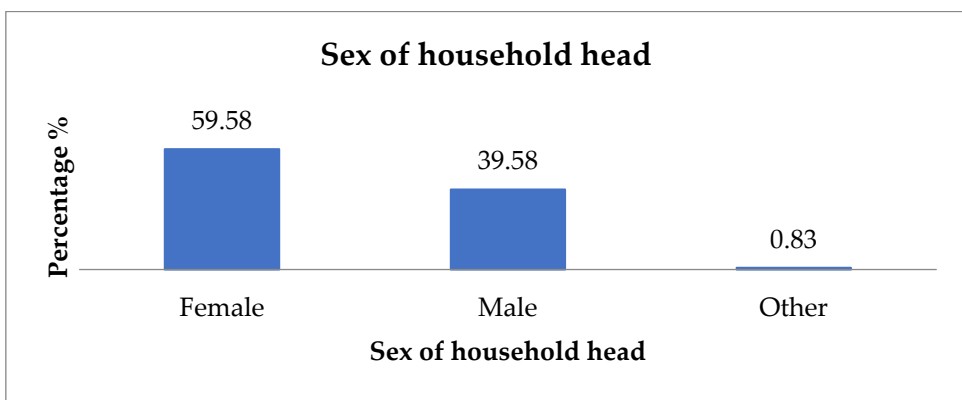

**Figure 5.** Sex of household heads [58]. **Source:** Authors' computation from survey data.

Figure 6 shows the distribution of households by sex and their vulnerability to drought in selected study areas in Western Cape province. The result shows that female-headed households are more vulnerable to drought than their male counterparts, with only a few of these households not vulnerable to drought. This result may be because the majority of the respondents in the study area were female. This is consistent with studies conducted by Sagnestam [64] and Mekuyie [65] in Nicaragua and southern Ethiopia, respectively, which found that female-headed households have a lower potential to adopt long-term, adaptive solutions than their male counterparts.

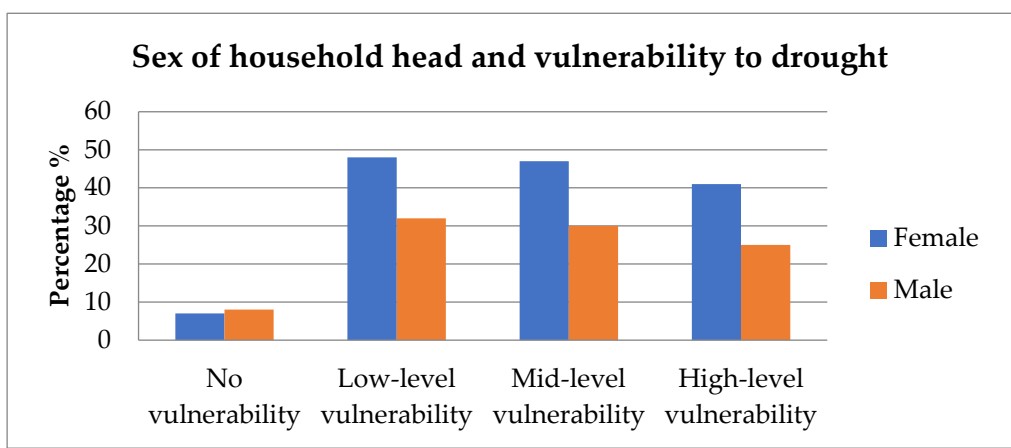

**Figure 6.** Sex of household heads and vulnerability to drought [58]. **Source:** Authors' computation from survey data.

Figure 7 shows the distribution of household heads by age group in the study area. There are four age categories, ranging from 18 years old to over 70 years old. The household head age range between 30 and 49 years is the highest among respondents from the selected study area at 39%, followed by 50–69 years at 34%, 18–29 years at 17%, and the lowest is between 70+ years at 10%. This result shows that the majority of the respondents are at an active age. This result is in line with StatsSA [56] and the Western Cape government [63], showing that 70+ years is the lowest in the province, with the majority being 0–69 years of age.

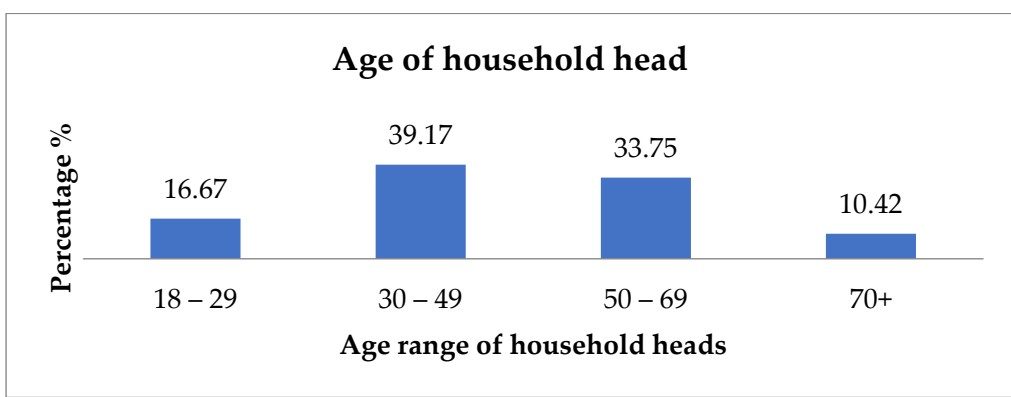

**Figure 7.** Age of household heads [58]. **Source:** Authors' computation from data.

Figure 8 shows the distribution of household heads by age group and their vulnerability to drought in the selected study areas of Western Cape province. Age is one of the determinants of vulnerability [66]. The result shows that the older the household head is, the more vulnerable the household is to drought. At 70+, there were no household heads in the study area that were not vulnerable to drought. According to Berbelet et al. [67], drought tends to increase older people's vulnerabilities, such as their mobility being reduced and their dependence on others increasing. In addition, older people might contribute to coping capacities in terms of their wealth of indigenous knowledge and experience [66]. However, they are the most vulnerable in that they may not have the ability to apply such knowledge without assistance from younger and abler people [68]. This age group is mature with respect to life issues in general.

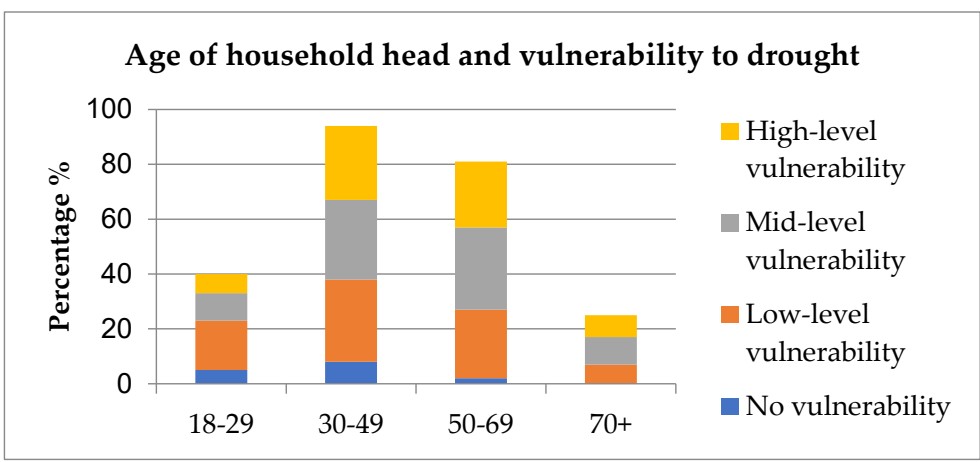

**Figure 8.** Age of household heads and vulnerability to drought [58]. **Source:** Authors' computation from data.

Figure 9 shows a summary of the household size of households in selected areas of Western Cape province. The mean household size is 4.5, while the minimum household size is 1, and the maximum is 17 household members. This information was necessary because the number of household members is likely to impact how much water is used in the household. Water consumption increases with household size [69].

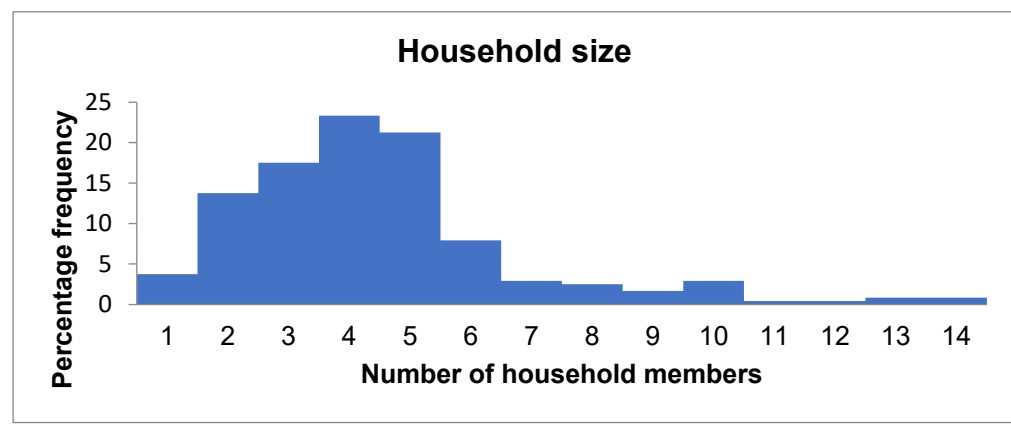

**Figure 9.** Distribution of household size [58]. **Source:** Authors' computation from data.

Figure 10 shows the distribution of households' awareness of water in the study area. It shows that the majority of the households were fully aware of the water restriction imposed by the authorities. The result also reveals that less than 9% of the households were not aware of the restrictions. According to Eid and Øyslebø [70], when awareness is raised, people can adapt to new social norms and engage in collective efforts to address an imminent environmental crisis. This is in line with a study by Matikinca et al. [69] that found households obtained information about the Day Zero communication campaign and rationing of water from a range of different sources, such as social media, posters, municipal workers, and community members.

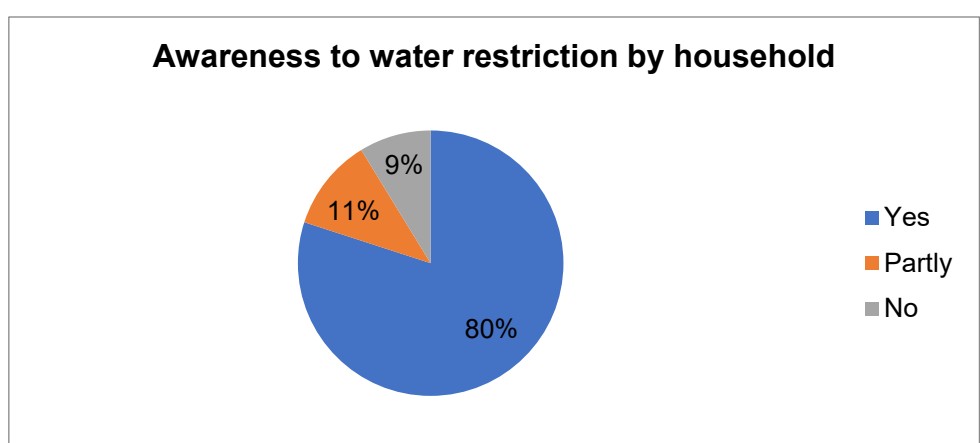

**Figure 10.** Distribution of households' awareness of water restrictions in selected areas of Western Cape province [58]. **Source:** Authors 'computation from data.

Table 2 shows the distribution of the households' perceptions and the public's awareness of the need to save water in the study area. The result shows that a majority of households agree that the public is aware of the need to save water. Awareness of water scarcity, personal responsibility, and the appropriate actions to take were shown to be important determinants of reducing water consumption [71]. There is huge potential in voluntary water conservation because the first step towards a household deciding to conserve water is that they understand the importance of water conservation and that they know what to do to reduce water demand [72]. The table also shows the distribution of the households' perceptions of the cooperation of the public with water restrictions in the study area. The result shows that about 59% of the respondents are positive that the public is cooperating with the water restriction imposed by the authorities. The result also shows that a reasonable percentage of the respondents strongly disagree, disagree, and are neutral that the public is cooperating with the water restriction imposed by the authorities. In 2019,

the water crisis was less imminent, but residents were encouraged to continue to preserve water and remain conscious of a potentially volatile environmental situation [70].

**Table 2.** Public awareness on the need to save water, public cooperation with water restrictions and authorities' drought management.

| | Strongly Agree | Agree | Neutral | Disagree | Strongly Disagree | Total |
|---|---|---|---|---|---|---|
| Public awareness on the need to save water | 37.92 | 42.92 | 9.17 | 5.42 | 4.58 | 100.00 |
| Public cooperation with water restrictions | 22.92 | 35.83 | 17.50 | 17.50 | 6.25 | 100.00 |
| Authorities on drought disaster | 10.42 | 31.67 | 23.33 | 22.92 | 11.67 | 100.00 |

Source: HSRC, 2019.

Table 2 also shows the distribution of the households' perceptions of the authorities' effectiveness in dealing with the drought situation in the study area. The result shows that about 42% of the respondents strongly agree or agree that the authorities were effective in dealing with the drought disaster. The result also shows that about 58% of the respondents strongly disagree, disagree, and are neutral that the authorities were effective in dealing with the drought disaster. The result shows that the authorities must do more to deal effectively with the drought disaster in the selected study areas. In South Africa, studies of past drought management have also indicated a trend of focusing on relief and emergency support, instead of implementing proactive policies [73].

Figure 11 shows the distribution of the households' water consumption if water decreased, stayed the same or increased over the last two years. The result shows that 59% of the respondents decreased their water consumption, 30% maintained the same water consumption, and about 12% increased their water consumption. According to Matikinca et al. [69], most households in CoCT changed the way they used water in their households, as the majority mentioned toilet flushing as the most common use of water in their households before the crisis and the introduction of greywater (bath water collected) for flushing under the water restrictions.

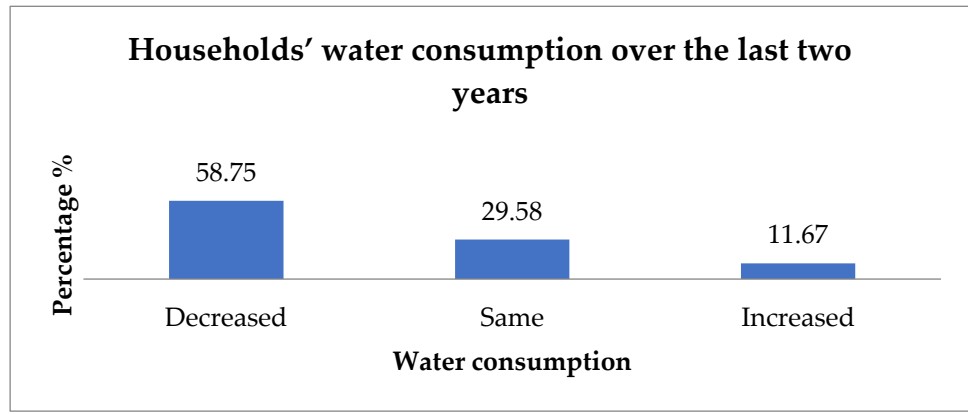

**Figure 11.** Households' water consumption over the last two years [58]. **Source:** Authors' computation from data.

Figure 12 shows the distribution of households' perception of the authorities communicating water restrictions in selected areas of Western Cape province. It shows that the majority of the households either agree or partly agree that the authorities communicated the water restrictions well. The result also reveals that about 20% of the households did not agree that the authorities communicated the water restrictions well. In a study by Matikinca et al. [69] and Eid and Øyslebø [70], some respondents indicated that they were frustrated because CoCT did not provide sufficient and clear information about restrictions.

They criticized the initial passive attitude from the authorities and emphasized how the government could have provided more information at an earlier point in time and found solutions to preserve water, while others noted that they were fine with water restrictions being ramped up, stating that the drought had put Cape Town in a very bad situation concerning the availability of water.

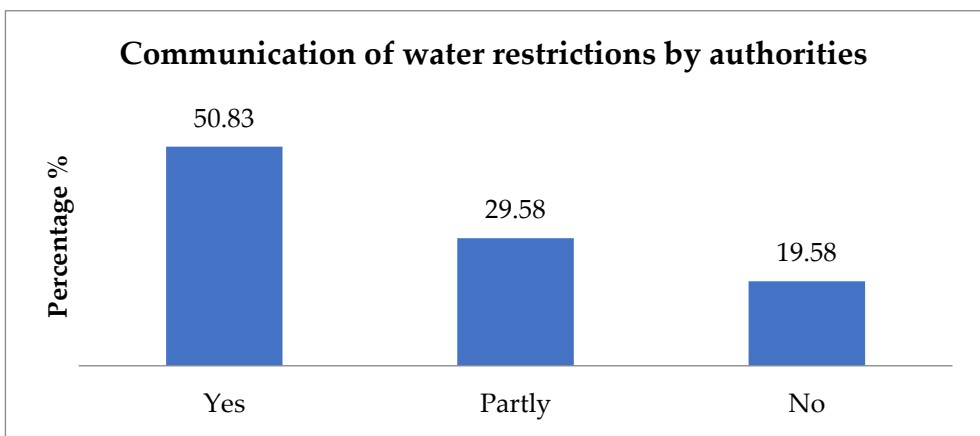

**Figure 12.** Communication of water restrictions by authorities [58]. **Source:** Authors' computation from data.

*4.2. Socio-Economic Determinants of Households' Vulnerability to Drought in Western Cape Province, South Africa*

Table 3 shows the logistic regression results showing determinants of households' vulnerability to drought in the Western Cape province, South Africa. The table shows that the age of the household head, communication of water restrictions by authorities, households' water consumption in the past two years, and public cooperation with water restrictions were statistically significant. The age of the household head is statistically significant and has a positive influence on the households' vulnerability level to drought. Assuming all other variables are constant, an increase in the age of the household head makes it more likely that the household will be vulnerable to drought. This result is in line with studies by Carter et al. [74] and Jimoh, Bikam, and Chikoore [75] that show the older the household head, the higher the vulnerability level of the household.

**Table 3.** Logistic regression results showing determinants of households' vulnerability to drought in Western Cape province.

| Dependent Variable: Household Vulnerability to Drought<br>Method: Ordered Logistic Regression<br>Iteration 0: Log Likelihood = −302.46686<br>Iteration 1: Log Likelihood = −288.95786<br>Iteration 2: Log Likelihood = −288.85282<br>Iteration 3: Log Likelihood = −288.85282<br>Iteration 4: Log Likelihood = −288.85282 | | |
|---|---|---|
| **Independent Variables (Socio-Economic Characteristics of the Households)** | **Coefficient** | **Std. Error** |
| Sex of head of household | 0.1592146 | 0.2372862 |
| Age of head of household | 0.3228623 ** | 0.1394568 |
| Household size | 0.0108975 | 0.0527702 |
| Communication of water restrictions by authorities | −0.4375439 *** | 0.1754546 |
| Household water consumption in the past two years | −0.5460676 *** | 0.1883137 |

**Table 3.** *Cont.*

| Dependent Variable: Household Vulnerability to Drought Method: Ordered Logistic Regression Iteration 0: Log Likelihood = −302.46686 Iteration 1: Log Likelihood = −288.95786 Iteration 2: Log Likelihood = −288.85282 Iteration 3: Log Likelihood = −288.85282 Iteration 4: Log Likelihood = −288.85282 | | |
|---|---|---|
| **Independent Variables (Socio-Economic Characteristics of the Households)** | **Coefficient** | **Std. Error** |
| Response by authorities to drought | −0.059939 | 0.1100499 |
| Public cooperation with water restrictions | 0.218828 ** | 0.1044304 |
| Household water saving methods | −0.0935807 | 0.0887855 |
| No. of observations | 240 | |
| Log likelihood | −288.85282 | |
| LR chi2(8) | 27.23 | |
| Prob > chi$^2$ | 30.8 | |
| Pseudo R$^2$ | 0.0450 | |

** Coefficients significant at 5%. *** Coefficients significant at 1%.

The communication of water restrictions by the authorities was statistically significant and had a negative influence on the households' vulnerability to drought in the study area. This shows that an improvement in the communication of water restrictions by the authorities reduces the likelihood of the household experiencing drought, assuming all other variables are held constant. The communication of water restrictions in CoCT has changed the way household members use water around their households [69]. The changes in their water use practices were mainly prompted by an increase in water restrictions and the Day Zero communication campaign. As a result of the authorities' communication of the water restrictions, households will be less vulnerable to drought because they were made aware of the situation in advance.

The household consumption of water in the past two years in the study area was statistically significant and has a negative influence on the households' vulnerability to drought. This shows that an improvement in the consumption of water in the past two years reduces the likelihood of the household experiencing a drought, assuming all other variables are held constant. In this study, an improvement in the consumption of water by the household is a reduction in the amount of water used by household members due to the restrictions by the authorities.

The final significant variable is public cooperation. Public cooperation was statistically significant and had a positive influence on the households' vulnerability to drought in the study area. With a lack of cooperation from the public on water usage, households are more likely to be vulnerable to drought, assuming all other variables are held constant. According to Alam's [76] study on drought adaptation, a household is less vulnerable to drought when they are more aware of the danger associated in not complying with limitations or other measures.

Table 4 shows the marginal effect of household vulnerability to drought in the study area. The table shows the marginal effect of having no vulnerability to drought. The age of the household head was statistically significant and has a negative influence on the households' ability to not be vulnerable to drought. Assuming all other variables are constant, an increase in the age of the household head reduces the likelihood of the household not being vulnerable to drought in the study area. Table 4 also shows that communication of water restrictions was statistically significant and had a positive influence on households' not being vulnerable to drought in the study area. It shows that an improvement in the

communication of water restrictions by authorities makes it more likely for the household not to be vulnerable to drought, assuming all other variables are constant. The table below also shows that the water consumption of the household in the past two years was statistically significant and has a positive influence on households' not being vulnerable to drought in the study area. If the household's water consumption in the past two years has been well managed (reduced usage), the household is more likely not to be vulnerable to drought, all other variables held constant. Finally, the table below shows that the public's cooperation with water restrictions was statistically significant and had a negative influence on the households' ability to not be vulnerable to drought. An improvement in public cooperation with water restrictions makes the household less likely to be vulnerable to drought, assuming all other variables are held constant.

**Table 4.** Marginal effect after logit (no vulnerability).

| Dependent Variable: Household Vulnerability to Drought Marginal Effects after logit Y = Pr(Household Vulnerability to Drought = 1) (Predict, Outcome(1)) = 0.05332566 | | |
| --- | --- | --- |
| **Independent Variables (Socio-Economic Characteristics of the Households)** | **Coefficient** | **Std. Error** |
| Age of head of household | −0.0606124 ** | 0.0268 |
| Communication of water restrictions by authorities | 0.0821421 *** | 0.03362 |
| Household water consumption in the past two years | 0.1025158 *** | 0.03681 |
| Public cooperation with water restrictions | −0.0410816 ** | 0.01995 |

** Coefficients significant at 5%. *** Coefficients significant at 1%.

Table 5 shows the marginal effect of household vulnerability to drought in the study area. The table shows the marginal effect of high-level vulnerability to drought. The age of the household head was statistically significant and has a positive influence on the households experiencing a high level of vulnerability to drought. This shows that an increase in the age of the household head is likely to cause the household to experience high-level vulnerability to drought, assuming all other variables are held constant. Table 5 also shows that communication of water restrictions was statistically significant and has a negative influence on households experiencing a high level of vulnerability to drought in the study area. It shows that an improvement in the communication of water restrictions by authorities makes it less likely for the household to experience high-level vulnerability to drought, assuming all other variables are held constant. The table below also shows that the water consumption of the household in the past two years was statistically significant and had a negative influence on the households' experiencing a high level of vulnerability to drought. If the household's water consumption in the past two years was well managed (reduced usage), the household is less likely to experience high-level vulnerability to drought, all other variables held constant. Finally, the table below shows that the public's cooperation with water restrictions was statistically significant and had a positive influence on the households' experiencing a high level of vulnerability to drought. The household is more vulnerable to drought if there is a lack of public cooperation with water restrictions.

**Table 5.** Marginal effect after logit (high-level vulnerability).

| Dependent Variable: Household Vulnerability to Drought Marginal Effects after logit Y = Pr(Household Vulnerability to Drought = 4) (Predict, Outcome(4)) = 0.05332566 | | |
|---|---|---|
| **Independent Variables (Socio-Economic Characteristics of the Households)** | **Coefficient** | **Std. Error** |
| Age of head of household | 0.0616107 ** | 0.0266 |
| Communication of water restrictions by authorities | −0.08344951 * | 0.03378 |
| Household water consumption in the past two years | −0.1042043 * | 0.0359 |
| Public cooperation with water restrictions | 0.0417582 ** | 0.02001 |

* Coefficients significant at 10%. ** Coefficients significant at 5%.

## 5. Conclusions and Recommendations

Drought has a broader influence on a household than just a decrease in rainfall. The impact manifests itself in a variety of ways, including economic impact, health, and food security, among others. As a result, the purpose of this study was to investigate the determinants of household vulnerability to drought in the Western Cape, South Africa. The study area was the CoCT and the West Coast District municipality (Clanwilliam and Piketberg). The level of vulnerability was classified into four categories: no vulnerability, low-level vulnerability, mid-level vulnerability, and high-level vulnerability. According to the study, about 94% of the households experienced vulnerability, with 28% of the households experiencing a high level of vulnerability to drought. Approximately 60% of the households in the study area were headed by females. The study found that female-headed households were more vulnerable to drought than their male counterparts. Household heads in the study area were mostly between the ages of 30 and 49, and between the ages of 50 and 69, respectively. The distribution of household heads by age and vulnerability to drought found that the older the head of the household, the more vulnerable the household is to drought.

The findings of the study also revealed that a sizeable number of the households were aware of the water restrictions imposed by the authorities. The majority of households feel that the public is aware of the need to save water since the authorities used various methods to communicate the Day Zero campaign, water restrictions, and the change in prices. The results also suggest that many of the households surveyed disagreed with the authorities' effectiveness in dealing with the drought disaster because the respondents felt frightened by the authorities' restrictions, tariffs, and fear of running out of water.

According to the results of the OLR, indicators such as the age of the household head, communication of water restrictions by authorities, household water consumption in the last two years, and public cooperation with water restrictions were statistically significant as to the households' vulnerability to drought in the Western Cape province, South Africa. The age of the household head and public cooperation with water restrictions were statistically significant and had a positive influence on the households' vulnerability to drought. The result shows that the older the head of the household, the more vulnerable the household is to drought. This is in line with the descriptive analysis of the age of the head of the household and the vulnerability to drought. The descriptive analysis shows that heads of households aged 70+ in the study area were very vulnerable to the drought. The communication of water restrictions by authorities and household water consumption in the last two years were also statistically significant but had a negative influence on the households' vulnerability to drought.

Based on the results of this study, the majority of the households in the study area were vulnerable to drought. The study further revealed that proactive measures should be taken to decrease the household's vulnerability to drought. Natural disasters such as drought are expected to increase due to climate change, and it is critical that humanitarian actions not only respond to the specific needs and vulnerabilities of weaker members of society, e.g., older citizens, female-headed households, and children, but also recognise and build on their capacities to contribute to humanitarian preparedness and response. Improved, proactive decision making and communication of water restrictions should be adopted to better prepare and educate households promptly. Different means of awareness, such as media campaigns through posters, radio, television, and short message service (SMS) may be used and, most importantly, the message should be communicated in the language the majority of the households in the study area understand. This conclusion implies that, despite the achievements made by the government and many stakeholders in the study area during the drought period, further implementations could be made in terms of communication to increase readiness for a period of water scarcity. The study also highlighted the necessity to educate households in the area on the benefits of saving water through reasonable reductions in water usage; this role may be played by various stakeholders, including sensitized household members. Furthermore, the study suggests sustainable water sources to ensure that water supply remains consistent in the face of climate change impacts such as a lack of rainfall, drought, or too much rain. Water reuse and desalination of water, for example, can be employed as sustainable water sources in the study area, which is bordered by the ocean. Water reuse may also be utilized as a sustainable source of water supply, reducing demand on main water resources such as surface and groundwater. Desalination of water, when combined with renewable energy, might also assist in supplying a sustainable source of water.

**Author Contributions:** Conceptualization, I.B.O. and T.M.B.; data curation, I.B.O. and T.M.B.; methodology, I.B.O. and T.M.B.; data analysis, I.B.O. and T.M.B.; writing—original draft preparation, I.B.O. and T.M.B.; writing—review and editing, I.B.O. and T.M.B.; supervision, I.B.O. and T.M.B. All authors have read and agreed to the published version of the manuscript.

**Funding:** This research received no external funding.

**Institutional Review Board Statement:** Not applicable.

**Informed Consent Statement:** Informed consent was obtained from all subjects involved in the study.

**Data Availability Statement:** Human Sciences Research Council (HSRC). Adaptation to the Western Cape drought of 2016-18 (WCD), 2019: Household and business survey. [Data set]. WCD 2019. Version 1.0. Pretoria South Africa: Human Sciences Research Council [distributor] 2020. http://doi.org/10.14749/1570105897.

**Acknowledgments:** The authors would like to thank HSRC for making the data accessible.

**Conflicts of Interest:** The authors declare that there is no conflict of interest.

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
