# Peer review of "Socioeconomic Determinants of Households’ Vulnerability to Drought in Western Cape, South Africa"

_sustainability, doi:10.3390/su14137582_

Round 1

Reviewer 1 Report

The authors carried out interesting research in the households’ vulnerability to drought in western cape, South Africa. The research is well presented.

Author Response

Dear Sir,

Attached is the rebuttal to the comments from Reviewer 1.

Reviewer 2 Report

Dear Professor,

I have reviewed the revise manuscript entitled “Socioeconomic Determinants of Households’ Vulnerability to Drought in Western Cape, South Africa. The content of the paper is interesting, and the presentation of the paper is also good. However, I am not expertized in the field of drought. It will be better if expertized reviewer in the field of drought give decision on this paper.

Thanking you

Author Response

Dear Sir,

Attached is the rebuttal to the comments from Reviewer 2.

Reviewer 3 Report

  1. Please add the research gap in the end of the Introduction.
  2. The paper does not demonstrate enough relevant literature, particularly to develop a research model in this paper. The references are very lacking. Please update the Literature (up to 2015)
  3. The authors have not presented the research discussion and result of the study clearly.
  4. The discussion should be done more thoroughly by connecting previous studies and the finding of the research

Author Response

The comments received from Reviewer 3 have been incorporated as part of the Editor's comments

Reviewer 4 Report

Socioeconomic Determinants of Households’ Vulnerability to Drought in Western Cape, South Africa

This paper examines the determinants of household vulnerability to drought in the Western Cape province, South Africa. Statistics was used to analyze households’ socioeconomic variables, the logit model was employed to analyze the factors contributing to households’ vulnerability to drought in the study area. The article deals with an important topic that can be used to improve public policies. However, this manuscript requires an important revision. The authors need to better discuss the results they found, in addition to improving the figures presented. These and other issues should be considered/clarified prior to acceptance.

  1. The introduction is too repetitive, and doesn't follow a line of reasoning. I suggest rewriting first defining the concept of drought, the impacts it causes, the changes in duration and intensities due to anthropization and finally the impacts on the study site.

  1. Linhas 45 - 46 apresenta o mesmo conceito mostrado nas Linhas 29 - 20.

  1. The importance of this study is not fully reflected. Please highlight the innovation in the introduction and the abstract. This article is intended to provide a reference for research on the drought impact on the population. However, it does not explore in the introduction what has been studied on the subject, or its originality.

  1. The very simple introduction. It does not present the state of the art of the studied subject. It does not show other types of analyzes for different locations.

  1. Drought

  1. First paragraph again repeats drought concepts presented in the introduction. Mainly on Lines 58 - 62.

  1. This topic doesn't contain information different from that presented in topic 1 (Introduction). I suggest incorporating this topic into the Introduction.

  1. Impact and social aspects of drought

  1. Please check the journal's formatting. You start by enumerating the topics and after topic 3, you don't enumerate any more.

e.g. Drought in Western Cape Province

  1. Figure 1 wasn't  called out in the text.

  1. Figure 1 needs a map of South Africa to show the reader who does not know the country where the study city is located, the way it is presented makes it difficult for those who do not know the region to find it.

  1. Please describe in more detail the characteristics of the study region.

  1. Figure 2 is similar to figure 1. It was not called in the text.

  1. The texts referring to figures 1 and 2 appear after them. Please correct.

  1. The data and methods used are not well described. They need to be more detailed, in particular, the Ordinal Logistic Regression method, so that their work can be reproduced.

Results and Discussion

In general, the results need to be further discussed. Authors often just describe the figures and do not discuss them. It needs to compare with the state of the art, with different studies in different locations, thus highlighting the results found by the authors.

  1. As with Figures 1 and 2, all results figures appear before the corresponding text. Also, you need to improve them.

  1. Figures 3, 4 and 5 What does the y-axis represent?

  1. Figure 4, the first column on the left (value of 25.42) corresponds to what?

  1. Figure 6 and 8. What do the y-axis values represent?

  1. Figure 7, 10 and 11 are similar to topic 15 .

Conclusion and Recommendations

  1. The authors repeat a lot of what has already been described throughout the text.

Author Response

The few comments received from Reviewer 4 have been incorporated as part of the Editor's comments

Round 2

Reviewer 3 Report

The revision has been done completely

Reviewer 4 Report

Socioeconomic Determinants of Households’ Vulnerability to Drought in Western Cape, South Africa

This paper examines the determinants of household vulnerability to drought in the Western Cape province, South Africa. Statistics was used to analyze households’ socioeconomic variables, the logit model was employed to analyze the factors contributing to households’ vulnerability to drought in the study area. The article deals with an important topic that can be used to improve public policies. The authors made a major revision of the manuscript following the suggestions suggested by the reviewers. Which significantly improved the understanding of the results shown. Soon I Accept this paper.